# FedDUAL: A Dual-Strategy with Adaptive Loss and Dynamic Aggregation for Mitigating Data Heterogeneity in Federated Learning

## Abstract

Federated Learning (FL) marks a transformative approach to distributed model training by combining locally optimized models from various clients into a unified global model. While FL preserves data privacy by eliminating centralized storage, it encounters significant challenges such as performance degradation, slower convergence, and reduced robustness of the global model due to the heterogeneity in client data distributions. Among the various forms of data heterogeneity, label skew emerges as a particularly formidable and prevalent issue, especially in domains such as image classification. To address these challenges, we begin with comprehensive experiments to pinpoint the underlying issues in the FL training process. Based on our findings, we then introduce an innovative dual-strategy approach designed to effectively resolve these issues. First, we introduce an adaptive loss function for client-side training, meticulously crafted to preserve previously acquired knowledge while maintaining an optimal equilibrium between local optimization and global model coherence. Secondly, we develop a dynamic aggregation strategy for aggregating client models at the server. This approach adapts to each client's unique learning patterns, effectively addressing the challenges of diverse data across the network. Our comprehensive evaluation, conducted across three diverse real-world datasets, coupled with theoretical convergence guarantees, demonstrates the superior efficacy of our method compared to several established state-of-the-art approaches. The code can be found at https://anonymous.4open.science/r/FedDUAL-B04F/README.md.

## 1 Introduction

Federated learning (FL) has revolutionized collaborative model training by enabling multiple clients to contribute to a global model without compromising the privacy of their local data (McMahan et al., 2017). This decentralized strategy avoids the need for sending data to a central server, thus maintaining data privacy. As the digital landscape evolves, with an increasing number of distributed data sources emerging from mobile devices, healthcare institutions, and Internet of Things (IoT) networks, FL has emerged as a pivotal solution for training sophisticated deep networks across geographically dispersed and heterogeneous environments (Bonawitz et al., 2016), Sahoo et al. (2024b), (Hu et al., 2024). However, a significant practical obstacle encountered during federated training is data heterogeneity in the form of skewness in labels and quantity of the data across various clients (Kairouz et al., 2021), (Li et al., 2020). Diverse user behaviors can lead to significant heterogeneity in the local data of different clients, leading to non-independent and identically distributed (non-IID) data. This variability can introduce biases in model training, leading to unstable convergence and potentially degrading the model's performance or making it counterproductive (Li et al., 2022), (Zhao et al., 2018). While FedAvg (McMahan et al., 2017) is effective and widely used, it often falls short in accuracy and convergence with static aggregation methods. These methods combine model updates from different clients in a fixed manner, failing to adapt to heterogeneous data distributions and client drift, as discussed in (Karimireddy et al., 2020).

Previous studies have addressed the issue of client drift by implementing penalties for deviations between client and server models (Li et al., 2020), (Li et al., 2021a), employing variance reduction techniques during client updates (Karimireddy et al., 2020), (Acar et al., 2021), or utilizing novel aggregation methods on the server side (Chen et al., 2023), (Chowdhury & Halder, 2024).

## 1.1 Motivation

Prior studies by Yashwanth et al. (2024), Hu et al. (2024) have demonstrated that in non-IID scenarios, federated models tend to converge to 'sharp minima', resulting in significant performance degradation and compromised generalizability. In this study, we investigate the root causes of this phenomenon and propose a novel solution to mitigate its effects. Our study begins with a detailed analysis of loss landscapes for FedAvg-trained models across IID and non-IID data distributions. Figure 1 visually depicts the loss landscapes of two models on the FMNIST dataset with systematic parameter perturbations. The model trained on IID data exhibits a notably smoother and wider valley in its loss landscape, suggesting greater robustness and better generalization. In contrast, the model trained on non-IID manifests sharper peaks and narrower valleys, indicating higher sensitivity to parameter variations and potential overfitting. These visualizations offer strong evidence that in the presence of non-IID data, the FedAvg algorithm achieves suboptimal generalization. Motivated by this observation, we investigate the underlying mechanisms by analyzing gradient norms to identify which parts of the neural network are most affected by data heterogeneity. Our findings, presented in Fig. 2, 6 reveal a notable pattern: in non-IID scenarios, the gradient norms of the final layers, including the classification layer, exhibit significant amplification compared to their IID counterparts. Such amplification leads to model instability, impedes convergence, and ultimately compromises the generalizability of the federated model. Our investigation suggests that effective federated training in non-IID environments necessitates targeted adjustments during server-side aggregation, particularly for these highly affected layers, to achieve performance comparable to IID settings.

This prompts one critical question: *Can static aggregation methods effectively address severe non-IID data distributions across clients while maintaining higher convergence, performance, and generalizability in federated models?* The answer is decidedly negative. Static aggregation methods inherently

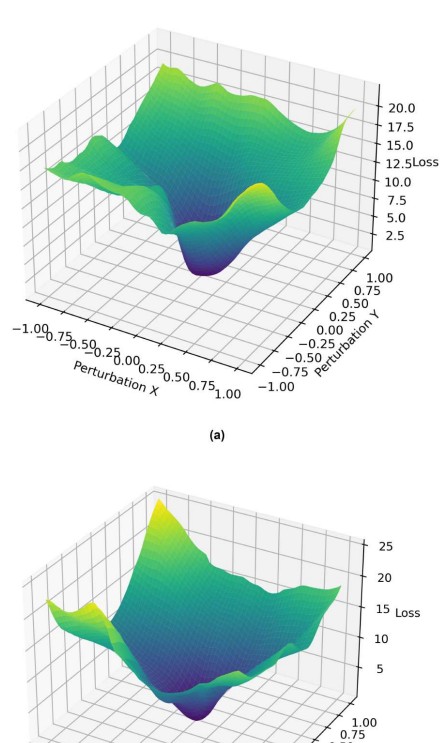

Figure 1: Visualization of the loss surface for the global model trained on the FM-NIST dataset using the FedAvg algorithm: (a) depicts the loss landscape when trained on IID data, while (b) illustrates the landscape for non-IID data distribution.

struggle with the dynamic heterogeneity present in federated networks, where adjusting parameters based on client distributions and performance in each communication round is crucial. Although incorporating predetermined parameters into the aggregation process may provide some partial mitigation, these methods fail to address the complex challenges posed by non-IID data distributions. A more dynamic and nuanced approach is necessary to effectively manage these multifaceted issues. To address this challenge, we apply dynamic aggregation to the model's final layers, where gradient norms fluctuate significantly in non-IID scenarios, while using traditional aggregation (FedAvg) for the lower layers. For dynamic aggregation, we leverage the concept of Wasserstein Barycenter (Agueh & Carlier, 2011), derived from optimal transport theory, to integrate client-specific learning behaviors in these affected layers. By minimizing discrepancies from non-IID data, the Wasserstein Barycenter helps to align gradients from diverse clients, offering precise

model updates. This approach ensures fair aggregation, adapts to data heterogeneity, reduces bias, and enhances robustness, ultimately leading to more stable model convergence and improved generalization.

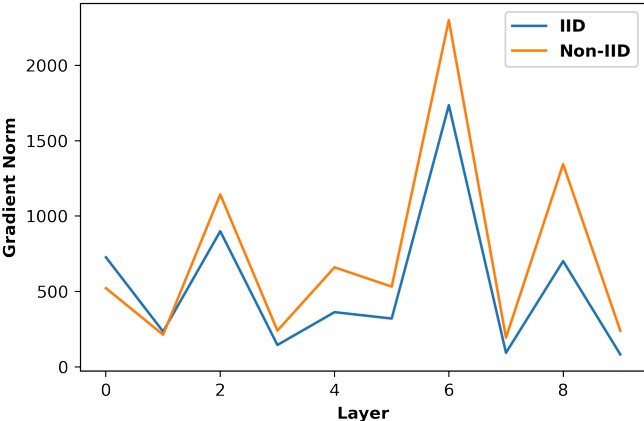

Figure 2: Comparison of gradient norms between models trained on IID and non-IID datasets using the FedAvg algorithm on the FMNIST dataset.

In addition to the server-side dynamic aggregation, we introduce an adaptive loss function for local training on the client side. This function allows clients to effectively explore the minima on their local datasets while preventing overfitting, thereby enhancing local optimization. Simultaneously, it preserves the global knowledge of the federated model, ensuring that the benefits from all participating clients are integrated. By incorporating a regularization parameter, $\beta$, the local loss function dynamically balances the trade-offs between local and global objectives. The contributions of this paper are as follows:

- We introduce FedDUAL, an innovative dual-strategy approach designed to effectively develop a robust and generalized federated model in highly heterogeneous data environments.

- We introduce an adaptive loss function for client-side training to balance the trade-offs between local and global objectives.

- Instead of straightforward server-side averaging, we propose a dynamic aggregation technique that uses Wasserstein Barycenter to reduce the effects of non-IID data by integrating the learning behaviors of participating clients.

- We conducted extensive experiments on three real-world datasets, demonstrating significant performance improvements over state-of-the-art methods and offering theoretical convergence guarantees for both convex and non-convex scenarios.

## 2 Related Work

The landscape of FL research has been significantly shaped by efforts to address data heterogeneity challenges, yielding a diverse array of innovative solutions. These approaches can be divided into three primary categories: (1) client drift mitigation strategies, which refine local client objectives to foster better alignment with the global model (Li et al., 2021a), (Karimireddy et al., 2020), (Acar et al., 2021), (Luo et al., 2021), (Li et al., 2023) (2) aggregation scheme optimization, aimed at enhancing server-side fusion of model updates (Hsu et al., 2019), (Lin et al., 2020), (Wang et al., 2020b), (Wang et al., 2020a) and (3) personalized FL, which tailors models to individual clients (Fallah et al., 2020), (Sattler et al., 2020), (Bui et al., 2019). Our research primarily focuses on two interconnected aspects of FL: mitigating client drift and optimizing server-side aggregation, and we will discuss the same in the literature review.

McMahan et al. (2017) introduced FL as an extension of local Stochastic Gradient Descent (SGD) (Stich, 2019), enabling increased local gradient updates on client devices before server synchronization and significantly reducing communication costs in identically distributed data settings. However, the method faces considerable obstacles when dealing with non-IID scenarios. Since then, various methods have emerged to address the challenge of data heterogeneity in FL (Li et al., 2019), (Yang et al., 2021), (Lin et al., 2018), (Hsu et al., 2019). FedProx (Li et al., 2020) incorporates a proximal regularization term to the optimization function to reduce model drift and addresses client stragglers. However, this term can also lead to local updates being biased towards the previous global model, which may result in misalignment between local and global optima. Building on previous work, Acar et al. (2021) introduced a dynamic regularization term to align local updates more closely with global model parameters, effectively reducing client drift caused by local model overfitting. Sun et al. (2023) further advanced the field with a momentum-based algorithm that accelerates convergence by combining global gradient descent with a locally adaptive optimizer. Similarly, several studies use variance reduction techniques, such as SCAFFOLD (Karimireddy et al., 2020). However, this approach often results in higher communication costs due to the transmission of additional control variates (Halgamuge et al., 2009). FedPVR (Li et al., 2023) addresses these limitations by reassessing FedAvg's performance on deep neural networks, uncovering substantial diversity in the final classification layers. By proposing a targeted variance reduction strategy focused solely on these final layers, FedPVR outperforms several benchmarks. MOON (Li et al., 2021a) introduces an innovative model-contrastive framework leveraging a contrastive loss to align local client representations with the global model, effectively mitigating client drift, and enhancing convergence, particularly in challenging non-IID environments. Luo et al. (2021) introduced CCVR (Classifier Calibration and Variance Reduction), which employs a classifier regularization and calibration method to enhance federated learning performance. CCVR's approach involves fine-tuning the classifier using virtual representations sampled from an approximated Gaussian mixture model. Shi et al. (2023) introduced a novel differentially private federated learning (DPFL) algorithm that integrates the Sharpness-Aware Minimization (SAM) optimizer to enhance stability and robustness against weight perturbations. By generating flatter loss landscapes and reducing the impact of differential privacy (DP) noise, it mitigates performance degradation and achieves state-of-the-art results, supported by theoretical analysis and rigorous privacy guarantees. Fanì et al. (2024) proposed FED3R, leveraging Ridge Regression on pretrained features to tackle non-IID data challenges, effectively mitigating client drift, enhancing convergence, and optimizing efficiency in cross-device settings.

Another line of research targets optimizing server-side aggregation in FL. For instance, Hsu et al. (2019) investigated the impact of non-IID data on visual classification by creating datasets with diverse distributions and found that increased data heterogeneity negatively affected performance, leading them to propose server momentum as a potential solution. FedNova (Wang et al., 2020b) addressed the problem of objective inconsistency due to client heterogeneity in federated optimization by introducing a normalized averaging technique, which resolves this inconsistency and ensures rapid error convergence. Addressing the limitations of traditional parameter averaging methods, Lin et al. (2020) introduced ensemble distillation for model fusion. This approach allows for the flexible aggregation of heterogeneous client models by training a central classifier on unlabeled data, using the outputs from the client models as guidance. FedMRL (Sahoo et al., 2024a) introduced a novel framework by using a loss function that promotes fairness among clients and employed a multi-agent reinforcement learning for personalized proximal terms , and a self-organizing map to dynamically adjust server-side weights during aggregation.

## 3 Methods and Materials

We consider a practical FL scenario with non-IID data distribution among $K$ independent clients, each with local training data $D_k(x, y)$, where $(x, y)$ denoting the data points. We initialize the global model weights $\theta_r^g$ and share it to the participating clients. The clients download the weighs from the server and train it using their local dataset $D_k(x, y)$. The updated model parameters $\theta_k^r$ from each client $k$ for $r^{th}$ communication round are uploaded to the server to aggregate into a global model $\theta_r^g$. Our objective is to develop a robust global model by collaboratively training local models across clients, even under varying heterogeneous conditions. To formalize, we define the optimal global model $\theta^*$ as follows:

$$\theta^* = \min_\theta F(\theta) = \frac{1}{K} \sum_{k=1}^{K} f_k(\theta) \tag{1}$$

where $f_k(\theta)$ is defined in Eq. 2.

$$f_k(\theta) = E_{(x,y)\sim D_k}[\ell(f_\theta(x), y)] \tag{2}$$

where $\theta$ represents the global model parameters, $f_\theta(x)$ is the model's prediction, and $\ell$ is the loss function.

### 3.0.1 Client Side Update.

At the beginning of each round $t$, the server randomly selects a subset $S_t \subset K$ of clients to participate in the federated training process and subsequently shares the current global model $\theta_r^g$ to these participating clients. Each client updates its local model by initializing with the global model parameters $(\theta_k^r = \theta_g^r)$ and then updates its local model by minimizing the local objective function. For local training, we have developed an adaptive objective function that balances local loss with the divergence between local and global models. The extent of this divergence is quantified using the Kullback-Leibler (KL) divergence (Csiszár, 1975), which effectively compares the probability distributions of the local model weights $p^k(w)$ with the global model weights $q(w)$. The KL divergence is mathematically defined in Eq. 4. To obtain the probability distributions of the local and global model weights, we first flatten the weights and then apply the softmax function. This process yields the desired probability distributions (p), as specified in Eq. 3.

$$p = \frac{\exp(\text{flatten weights})}{\sum \exp(\text{flatten weights})} \tag{3}$$

$$D_{\mathrm{KL}}(p^k \| q) = \sum_i p_i^k(w) \log\left(\frac{p_i^k(w)}{q_i(w)}\right) \tag{4}$$

where $p_i^k$ and $q_i$ are the probabilities associated with the $i^{th}$ component of the weight vectors. The local model must excel on local data while maintaining alignment with the global model to enhance overall generalization. This balance between minimizing local loss and aligning with the global model is defined as local adaptive function $\mathcal{L}_{adaptive}^k$ in Eq. 5.

$$\mathcal{L}_{adaptive}^k = (1 - \beta) * \mathcal{L}_{\mathrm{local}}^k + \beta * D_{\mathrm{KL}}(p^k \| q) \tag{5}$$

where $\mathcal{L}_{\mathrm{local}}^k$ is cross-entropy loss for $k^{th}$ client and $\beta$ is a regularization parameter and should be adaptive to account for the performance discrepancy between the local and global models. When the local model substantially outperforms the global model, $\beta$ should increase to enforce greater alignment. Conversely, if the models perform similarly, $\beta$ should decrease, allowing the local model to focus more on local optimization. The definition of $\beta$ is given in Eq. 6.

$$\beta = \sigma(\mathcal{A}_{\mathrm{local}}^k - \mathcal{A}_{\mathrm{global}}^k) \tag{6}$$

where $\sigma$ is the sigmoid function, $\mathcal{A}_{\mathrm{local}}^k$ represents the local model accuracy, and $\mathcal{A}_{\mathrm{global}}^k$ is the global model accuracy for client $k$. We calculated the global model's accuracy $\mathcal{A}_{\mathrm{global}}^k$ for client $k$ by evaluating it on the training data of client $k$ prior to performing local updates in the current round. Incorporating the adaptive parameter $\beta$ in Eq. 5, the adaptive loss function for client $k$ is represented in Eq. 7.

$$\mathcal{L}_{\mathrm{adaptive}}^k = (1 - (\sigma(\mathcal{A}_{\mathrm{local}}^k - \mathcal{A}_{\mathrm{global}}^k)) * \mathcal{L}_{\mathrm{local}}^k$$
$$+ \sigma(\mathcal{A}_{\mathrm{local}}^k - \mathcal{A}_{\mathrm{global}}) * D_{\mathrm{KL}}(p^k \| q) \tag{7}$$

After defining the adaptive loss function for each client, we optimize the local model parameters using stochastic gradient descent (SGD). The gradient update for the local model weights $w_k$ based on the adaptive loss function is given in Eq. 8.

$$w_k^{t+1} = w_k^t - \eta \nabla_w \mathcal{L}_{\text{adaptive}}^k(w_k^t) \tag{8}$$

where $\eta$ is the local learning rate. Expanding the gradient term $\nabla_w \mathcal{L}_{\text{adaptive}}^k(w_k^t)$, we obtain Eq. 9.

$$\nabla_w \mathcal{L}_{\text{adaptive}}^k(w_k^t) = (1 - (\sigma(\mathcal{A}_{\text{local}}^k - \mathcal{A}_{\text{global}}^k)) \nabla_w \mathcal{L}_{\text{local}}^k(w_k^t) + \\ \sigma(\mathcal{A}_{\text{local}}^k - \mathcal{A}_{\text{global}}) \nabla_w D_{\text{KL}}(p^k \| q) \tag{9}$$

The KL divergence term, $\sigma(\mathcal{A}_{\text{local}}^k - \mathcal{A}_{\text{global}}^k) \nabla_w D_{\text{KL}}(p^k \| q)$ in Eq. 9, acts as a regularizer to keep the local model gradients aligned with the global model gradients, thereby preserving model coherence despite non-IID data. By dynamically adjusting the regularization parameter $\beta$ based on the performance difference between local and global models, the adaptive loss function enhances the alignment of local models with the global model, thereby improving generalization and performance in non-IID scenarios. When local performance is less compared to global, the regularization term $\beta$ amplifies the focus on local optimization (first term in Eq. 5), enabling better-performing clients (that aligns well with the global model) to explore local optima more effectively. Conversely, if the global model performs worse, this term shifts the emphasis towards aligning with the global model (second term in Eq. 5), thereby supporting clients that are struggling by incorporating global knowledge.

### 3.0.2 Server Side Update.

After obtaining the weights from the participating clients at round $t$, the server calculates the Wasserstein Barycenter to effectively aggregate the weights of the last layers of the client models. Computing exact Wasserstein Barycenter can be computationally expensive, so we have approximated it using the Sinkhorn-Knopp (Knight, 2008) algorithm for efficient computation. We consider the local model weights as distributions and assign equal importance to each client in the computation of the Wasserstein Barycenter ($\bar{\mu}$). This barycenter represents the distribution that minimizes the sum of Wasserstein distances to the individual client gradient distributions, as formally defined in Eq. 10.

$$\hat{\mu} = \arg\min_{\nu} \sum_{k=1}^{K} \lambda_k W(\mu_k, \nu) \tag{10}$$

where $\lambda_k$ are weights corresponding to the importance or reliability of the client $k$. The Wasserstein distance $W(\mu_k, \mu_j)$ between two gradient distributions $\mu_k$ and $\mu_j$ of clients $j$ and $k$ is defined in Eq. 11.

$$W(\mu_k, \mu_j) = \left( \inf_{\gamma \in \Gamma(\mu_k, \mu_j)} \int_{\mathcal{X} \times \mathcal{X}} d(x, y)^p \, d\gamma(x, y) \right)^{1/p} \tag{11}$$

where $\Gamma(\mu_k, \mu_j)$ denotes the set of all couplings (or joint distributions) $\gamma$ on $\mathcal{X} \times \mathcal{X}$ with marginals $\mu_k$ and $\mu_j$ respectively, and $d(x, y)$ is the distance between points $x$ and $y$ in the metric space $\mathcal{X}$. After that, we use Sinkhorn-Knopp algorithm to calculate the Wasserstein Barycenter.

This barycenter is computed iteratively, starting by calculating a scaling factor $\gamma$ using Eq. 12, followed by Eq. 13.

$$\gamma = \exp\left( -\frac{W(\bar{p}, p_i)}{\epsilon} \right) \tag{12}$$

$$\bar{p}_{\text{new}} = \frac{\sum_i \lambda_i \gamma p_i}{\sum_i \lambda_i \gamma_i} \tag{13}$$

where $\bar{p}$ is the current estimate of the barycenter, $p_i$ refers to the $i^{th}$ client's gradient distribution, $\epsilon$ is a small positive constant, and the iterations continue until convergence. After few iterations, we get the Wasserstein barycenter that is used to update the global model weights. We update the the global model weights for the last layers by substracting them from the calculated Wasserstein barycenter for effectively aggregating the updates from the last layers. The algorithm of proposed method FedDUAL is given in the Algorithm 1. The proof of the convergence for both convex and non-convex settings for the proposed method can be found in the Section A of the Appendix.

---

**Algorithm 1** FedDUAL

---

1: **Input:** Number of clients $K$, Number of communication rounds $T$, and Global model $\mathcal{G}$.
2: **Output:** Trained global model $\mathcal{G}^*$.
3: Define a mask $e \in \{0,1\}^d$, where $e_j = 1$ for the last few layers and 0 for the rest layers.
4: Let $S_{naive} = \{j : e_j = 0\}$ and $S_{dynamic} = \{j : e_j = 1\}$.
5: **Initialize** global model weights $\theta^g$
6: **for** $t = 1$ to $T$ **do**
7:     Sample a subset of clients $\mathcal{S}_t \subseteq \{1, \ldots, K\}$
8:     Initialize lists: local model weights $\mathcal{W} \leftarrow []$, gradients $\Delta \leftarrow []$
9:     **for** each client $k \in \mathcal{S}_t$ **do**
10:         Initialize local model $\mathcal{M}_k$ with global weights $\theta^g$.
11:         Train $\mathcal{M}_k$ on local dataset $\mathcal{D}_k$ using adaptive loss function defined in Eq. 7.
12:         $\mathcal{W} \leftarrow \mathcal{W} \cup \{\theta_k\}$                                     ▷ Store local model weights $\theta_k$
13:         Compute gradients $\nabla_k$ for $\mathcal{M}_k$
14:         $\Delta \leftarrow \Delta \cup \{\nabla_k\}$                                        ▷ Store gradients $\nabla_k$
15:     **end for**
16:     **for** $j \in \{1, \ldots, d\}$ **do**
17:         **if** $e_j = 1$ **then**                                      ▷ Layer belongs to $S_{dynamic}$
18:             Extract last layers' gradients $\{\nabla_k[j]\}$ from $\Delta$
19:             Compute Wasserstein Barycenter of last layer $j$ gradients $\bar{\nabla}_j$
20:             Update global model's last layer $j$ weights $\theta^g[j] \leftarrow \theta^g[j] - \bar{\nabla}_j$
21:         **else**                                             ▷ Layer belongs to $S_{naive}$
22:             Perform Federated Averaging for layer $j$:
23:             $\theta^g[j] \leftarrow \frac{1}{|\mathcal{S}_t|} \sum_{k \in \mathcal{S}_t} \theta_k[j]$
24:         **end if**
25:     **end for**
26: **end for**
27: $\mathcal{G}^* \leftarrow \theta^g$                                                     ▷ Final trained global model

---

Table 1: Top-1 accuracy (%) on CIFAR10, CIFAR100, and FMNIST datasets. The values in bold represent the highest accuracy achieved. '*' denotes algorithms that failed to achieve convergence.

| | CIFAR10 | CIFAR100 | FMNIST |
|---|---|---|---|
| FedAvg | $46.68 \pm 0.25$ | $26.88 \pm 0.18$ | $81.70 \pm 0.20$ |
| FedProx | $47.58 \pm 0.30$ | $26.89 \pm 0.22$ | $80.54 \pm 0.28$ |
| FedNova | $48.44 \pm 0.35$ | * | * |
| FedBN | * | $26.88 \pm 0.19$ | $81.36 \pm 0.23$ |
| FedDyn | $43.97 \pm 0.40$ | $18.27 \pm 0.32$ | $71.86 \pm 0.45$ |
| MOON | $46.57 \pm 0.28$ | $28.50 \pm 0.25$ | $80.09 \pm 0.27$ |
| SCAFFOLD | * | * | * |
| FedPVR | $42.26 \pm 0.42$ | $23.78 \pm 0.31$ | $80.32 \pm 0.33$ |
| **Proposed** | $\mathbf{48.70 \pm 0.20}$ | $\mathbf{29.15 \pm 0.24}$ | $\mathbf{81.99 \pm 0.21}$ |

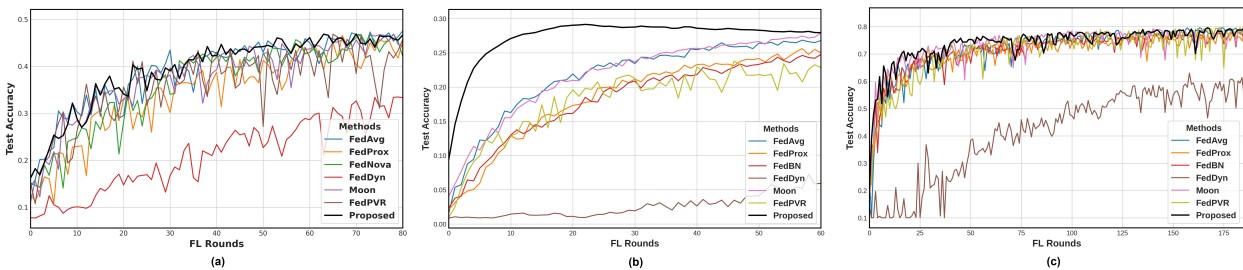

Figure 3: Learning curves comparing the proposed method with baselines across various datasets: (a) CIFAR-10, (b) CIFAR-100, and (c) FMNIST.

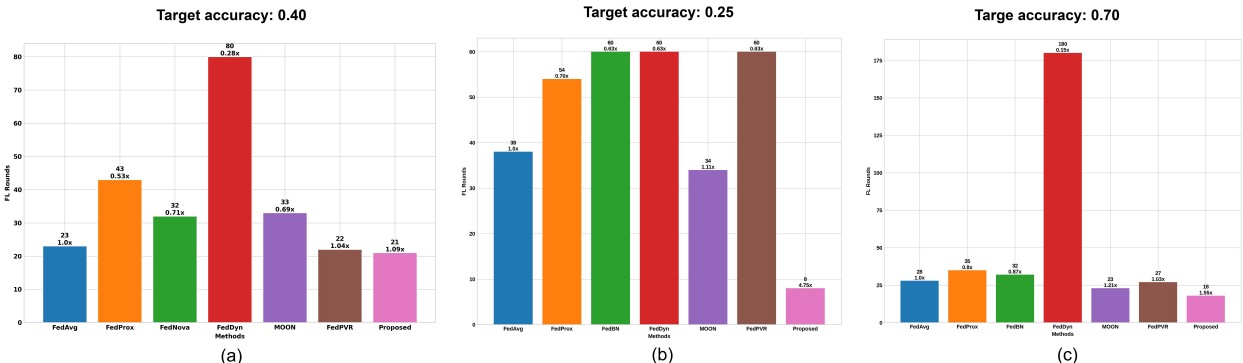

Figure 4: Number of FL rounds required to reach the target accuracy for the proposed method and other baselines on different datasets: (a) CIFAR-10, (b) CIFAR-100, and (c) FMNIST.

## 4 Convergence Proof

**Notation**

Let $\mathcal{W}$ denote the parameter space, $f_i : \mathcal{W} \to \mathbb{R}$ be the loss function for the $i^{th}$ client, and $F : \mathcal{W} \to \mathbb{R}$ be the global objective function.

**Assumptions**

**1. Lipschitz Smoothness:** For all $i$, $f_i$ is L-smooth such that it follows inequality 14:

$$\|\nabla f_i(w) - \nabla f_i(w')\| \le L\|w - w'\|, \forall w, w' \in \mathcal{W}. \tag{14}$$

**2. Bounded Variance:** The variance of stochastic gradients is bounded by inequality 15:

$$\mathbb{E}\|\nabla f_i(w, \xi) - \nabla f_i(w)\|^2 \le \sigma^2, \forall w \in \mathcal{W}. \tag{15}$$

where $\xi$ represents the random sampling of data.

**3. Bounded Norm:** We assume that the gradients of the local objective functions are uniformly bounded, i.e., there exists a constant $G > 0$ such that for all clients $i$ and for all model parameters $w$, the following holds:

$$\|\nabla F_i(w)\| \le G^2, \quad \forall i \in [N], \forall w \in \mathbb{R}^d. \tag{16}$$

**Theorem 1:** If the above assumptions hold, then our proposed FedDUAL achieves below rates in convex and non-convex settings.

**Convex setting:**

$$F(\bar{w}_T) - F(w^*) \leq \frac{\|w_0 - w^*\|^2}{2\eta T\sqrt{T}} + \frac{F(w_0) - F(w^*)}{2T} + \frac{L\eta G^2\sqrt{T}}{2T}.$$

**Non-Convex setting:**

$$\frac{1}{T}\sum_{t=0}^{T-1}\mathbb{E}\|\nabla F(w_t)\|^2 \leq \sqrt{\frac{2LG^2(F(w_0) - F(w^*))}{T}}$$

where G is the bound on the true gradient norm, L is the Lipschitz constant, and T is the number of communication rounds. The complete proof can be found in Section A of the Appendix and the discussion of obtained rates and comparison with existing methods can be found in Section B of the Appendix.

## 5 Experimental Results

### 5.1 Experimental Setup

To assess the effectiveness of the proposed FedDUAL approach, we conducted extensive experiments using three widely recognized classification benchmarks: CIFAR10 (Krizhevsky et al., 2009), CIFAR100 (Krizhevsky, 2009), and FMNIST (Xiao et al., 2017). To simulate real-world non-IID data distributions, we employed a client-wise partitioning strategy based on the Dirichlet distribution (Hsu et al., 2019). This distribution is governed by a concentration parameter $\alpha$, which controls the degree of data heterogeneity among clients. Lower $\alpha$ values result in more skewed data distributions, closely mimicking uneven data partitions. In all experiments, we set $\alpha = 0.01$ to simulate severe data heterogeneity, closely approximating real-world conditions. Throughout the communication rounds, each client retains a fixed local data partition. To evaluate the global model's classification performance, we use a separate test dataset maintained at the server, which remains unseen during training. For our experiments, we used LeNet (LeCun et al., 1998) for FMNIST dataset and a pre-trained VGG16 (Simonyan & Zisserman, 2015) for CIFAR-10 and CIFAR-100 dataset, following the methodology outlined in (Hu et al., 2024). We applied the proposed dynamic aggregation mechanism only to the last two layers of these models. Our setup involved 100 clients, with 10% randomly sampled per communication round, and a batch size of 32. Each client performed three local epochs of model updates. We have computed each result three times with different seed values and reported the mean value with standard deviation. To determine the optimal client learning rate for each experiment, we conducted a grid search over $0.05, 0.01, 0.2, 0.3$. For the baseline FedProx, we tested proximal values of $0.001, 0.1, 0.4, 0.7$ to find the optimal setting, and for FedNova, we evaluated proximal SGD values from $0.001, 0.003, 0.05, 0.1$, following the recommendations in Li et al. (2024). Across all experiments, we used the Adam optimizer for consistency. We have run each algorithm three times and reported the average outcome. The experimental setup utilized an NVIDIA Quadro RTX 4000 GPU boasting 40GB of memory. The implementation was crafted using Python [1], leveraging the TensorFlow framework [2] utilizing Windows 11.

### 5.2 Comparison with the State-of-the-art Methods

#### 5.2.1 Baseline.

We evaluate the proposed FedDUAL method against eight notable state-of-the-art (SOTA) FL baselines, including FedAvg (McMahan et al., 2017), FedProx (Li et al., 2020), FedNova (Wang et al., 2020b), SCAFFOLD (Karimireddy et al., 2020), FedBN (Li et al., 2021b), FedDyn (Acar et al., 2021), MOON (Li et al., 2021a) and FedPVR (Li et al., 2023).

---

[1]https://www.python.org/
[2]https://www.tensorflow.org/

### 5.2.2  Comparison of Accuracy.

The results, detailed in Table 1, reveal that many recent FL methods often fall short compared to the standard FedAvg baseline. In contrast, our proposed method consistently achieves SOTA performance, surpassing FedAvg along with other baselines across all evaluated scenarios. Furthermore, our approach exhibits remarkable adaptability across diverse datasets. Unlike some algorithms that excel on specific datasets but falter on others, the proposed FedDUAL consistently outperforms baselines across a wide range of data environments. This improvement suggests that our method addresses fundamental challenges in FL, potentially offering a more generalizable solution to the issues posed by data heterogeneity in federated settings. We also observed that FedNova, FedBN, and Scaffold did not perform effectively in our experimental setup.

### 5.2.3  Comparison of Convergence.

Figure 3 compares the learning curves of our method with baselines, while Fig. **??** in the Appendix includes the corresponding curves with error bars. The results demonstrate a consistent pattern: our approach exhibits accelerated learning and achieves superior accuracy across all tested scenarios. While the communication rounds vary by dataset, we observe that model performance reaches a saturation point by the conclusion of these rounds. Notably, our method not only attains a more robust final model but also displays markedly faster convergence across all datasets examined. This effectiveness is further highlighted in Fig. 4, where it consistently reaches target accuracy with far fewer communication rounds compared to baseline approaches.

### 5.3  Validation of the Motivation

To substantiate our claim that the proposed method yields models in flatter loss landscapes compared to FedAvg, we conducted a comparative analysis. Using VGG-16 models trained on the FMNIST dataset under non-IID conditions ($\alpha = 0.01$), we visualized their respective loss landscapes following the approach outlined in Li et al. (2018). Figure 7 in the Appendix depicts these landscapes, with each model centrally located within its respective terrain. The visualization reveals that our proposed method situates the model in a notably flatter region compared to FedAvg. This finding supports our assertion that our approach guides federated training towards more stable and generalizable solutions, characterized by flatter loss landscapes. The performance improvement of the proposed models stems from two key innovations: a Wasserstein Barycenter-based aggregation for final layer gradients, mitigating client drift in heterogeneous data environments, and an adaptive loss function balancing local optimization with global consistency during client training. This synergistic approach preserves global knowledge while promoting client-specific optimization, addressing fundamental FL challenges.

## 6  Ablation Study

In our ablation study, we performed all experiments on the FMNIST dataset with $\alpha = 0.01$. The study comprised four types of experiments: (1) performance analysis of the individual modules, (2) assessment of the impact of dynamic aggregation across different neural network layers, (3) hyperparameter analysis, and (4) evaluation of various levels of data heterogeneity.

### 6.0.1  Performance Analysis of Individual Modules

To assess the effectiveness of the proposed adaptive loss and dynamic aggregation techniques, we conducted three ablation experiments across FMNIST, CIFAR-10, and CIFAR-100. The results for FMNIST are shown in Table 2, while those for CIFAR-10 and CIFAR-100 are reported in Table 3 and Table 4 in the Appendix, respectively. In the first experiment, we employed only the adaptive loss alongside standard server-side aggregation. Notably, this configuration underperforms FedAvg across all datasets, indicating that adaptive loss alone cannot effectively address data heterogeneity—likely due to its limited capacity to improve generalization despite fostering local-global alignment. The second experiment implemented our dynamic aggregation technique at the server, while retaining the conventional cross-entropy loss function

Table 2: Ablation study results for the proposed FedDUAL method on the FMNIST dataset.

| Adaptive Loss | Dynamic Aggregation | Accuracy (%) |
|:---:|:---:|:---:|
| ✓ | ✗ | $80.70 \pm 0.22$ |
| ✗ | ✓ | $80.91 \pm 0.25$ |
| ✓ | ✓ | $\mathbf{81.99 \pm 0.18}$ |

locally. Finally, the third experiment combined both proposed methods: the adaptive loss function and the dynamic aggregation technique. As evidenced by Table 2, the integration of both proposed approaches in the third experiment yielded the highest accuracy, highlighting the impact of our dual strategy on model performance. The learning curves for these experiments using FMNIST dataset are illustrated in Fig. 8 of the Appendix.

### 6.0.2 Impact of Dynamic Aggregation on Different Network Layers

To substantiate our decision to apply dynamic aggregation technique selectively to last layers, we examined its impact across various layers of the neural network. Our earlier findings highlighted that data heterogeneity primarily affects last layers of the network. Figure 5 illustrates that random utilization of the dynamic aggregation to all layers diminishes performance. Conversely, targeted implementation on layers proximal to the classifier yielded optimal accuracy and convergence. These outcomes validate our hypothesis and demonstrate the method's efficacy in mitigating heterogeneity-induced issues. By focusing our dynamic aggregation technique on the most susceptible layers, we directly address the core challenge of data heterogeneity in federated training, resulting in enhanced model performance and faster convergence.

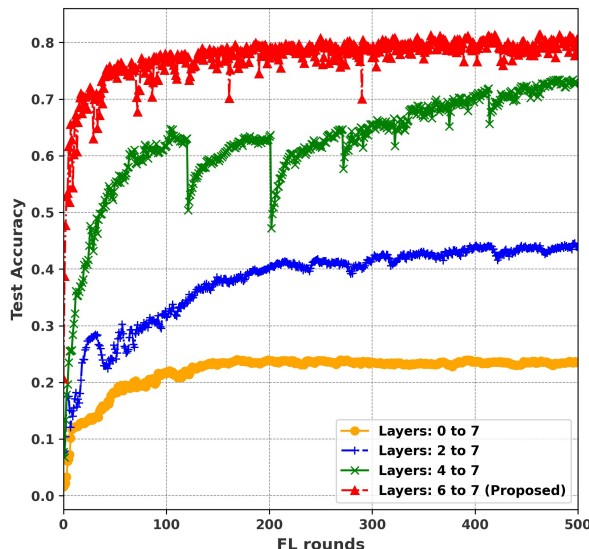

Figure 5: Illustration of the Dynamic aggregation method applied across various layers of the neural network.

### 6.0.3 Hyperparameter Analysis

In the proposed architecture, there are two key hyperparameters to consider: the scaling factor ($\gamma$) and the number of iterations used to compute the Wasserstein Barycenter. The proposed FedDUAL approach utilizes dynamic server-side aggregation by applying the Wasserstein Barycenter concept to combine the weights of the final layers from local models. This iterative process involves a small positive constant ($\epsilon$) to determine the scaling factor ($\gamma$). To optimize performance, we conducted two sets of experiments on the FMNIST dataset with $\alpha = 0.01$, each exploring a range of values for these crucial hyperparameters. The hyperparameter $\epsilon$ influences the sensitivity of the barycenter calculation to variations in Wasserstein distance. A smaller $\epsilon$ makes the barycenter more responsive to differences in Wasserstein distance, while a larger $\epsilon$ diminishes this sensitivity. This impacts how the barycenter integrates each distribution according to its distance from the current estimate. During the iterative update of the barycenter, $\epsilon$ affects the scaling factor $\gamma$ applied to each distribution. An excessively small $\epsilon$ can result in slow or potentially non-existent convergence due to minimal scaling factor, whereas a too-large $\epsilon$ may cause oversmoothing, reducing the barycenter's effectiveness in accurately representing the distributions. For this setting we have fixed the number of iterations to compute Wasserstein Barycenter as 150. Figure 9 in the Appendix shows test accuracy across different $\epsilon$ values, indicating that larger $\epsilon$ can degrade performance or hinder convergence. Figure 10 in the Appendix presents the corresponding learning curves for these settings. The number of

iterations in the Wasserstein Barycenter function is another critical hyperparameter that affects both the accuracy and efficiency of the barycenter computation. Generally, more iterations enhance convergence and accuracy, ensuring that the barycenter more closely approximates the optimal value. However, increasing the number of iterations also prolongs computation time, necessitating a balance between accuracy and efficiency. Finding the optimal number of iterations involves a trade-off: too few iterations may result in suboptimal outcomes, while too many can yield diminishing returns in accuracy. To achieve the best performance, begin with a reasonable default value, monitor convergence by observing changes in the barycenter, and adjust iteratively based on empirical results and available computational resources. For this setting, we fixed the *epsilon* value as 0.0001, which yields the highest results in previous experiment. Figure 11 illustrates the test accuracy for different values of iterations to calculate Wasserstein Barycenter, suggesting that larger iterations may adversely affect performance. Figure 12 presents the corresponding learning curves for these settings. From both experiments, we observe that the highest performance is achieved with $\epsilon = 0.00001$ and 150 iterations. Therefore, to optimize performance, it is advisable to set $\epsilon$ to a smaller value while keeping the number of iterations between 100 and 150.

### 6.0.4 Experiment on Different Level of Data Heterogeneity

Figure 13 illustrates the accuracy of the proposed method and various baselines across different levels of data heterogeneity on the FMNIST dataset. In this context, heterogeneity is quantified by $\alpha$, with lower values indicating greater data heterogeneity. The results show that as $\alpha$ decreases, the test accuracy for all models increases, because data heterogeneity among clients is decreased. Remarkably, the proposed method consistently achieves the highest test accuracy and exhibits the slowest performance decline compared to other algorithms, demonstrating superior performance of the proposed method on varying degrees of non-IID data partitioning. The learning curve is presented in Fig. 14.

## 7 Limitation and Future Work

While the proposed FedDUAL framework achieves superior performance and faster convergence under severe data heterogeneity, where several state-of-the-art methods such as FedNova, FedBN, and SCAFFOLD fail to converge, it introduces higher computational cost at the server due to the iterative calculation of the Wasserstein Barycenter. On the client side, FedDUAL remains lightweight and is also more communication-efficient than methods like SCAFFOLD and FedPVR, as it only requires transmitting model updates without additional control variates. Although this extra computational overhead is limited to the server, which typically has sufficient resources, future research will focus on reducing the computational burden of the Wasserstein Barycenter calculation by developing more efficient algorithms, aiming to maintain or even improve the performance of the proposed framework. Moreover, FedDUAL is inherently compatible with standard privacy-preserving techniques. In particular, its adaptive loss and dynamic aggregation can be seamlessly integrated with Differential Privacy by adding noise to client updates prior to aggregation, without modifying the core design. Investigating this integration with formal privacy guarantees remains an important direction for future research.

## 8 Conclusion

This research presents a novel approach to address the challenges posed by data heterogeneity among clients in the federated approach. We systematically analyze the factors contributing to federated model performance degradation under severe data heterogeneity and propose an architecture incorporating dual-strategy innovations. First, we implement an adaptive loss function for client-side training. Second, we create a dynamic aggregation strategy for server side aggregation, tailored to client-specific learning behaviors. The proposed FedDUAL effectively overcomes the challenges of heterogeneous data, outperforming eight SOTA baselines. It demonstrates faster convergence and consistently improved performance, making it an excellent solution for large-scale FL applications in real-world scenarios. Our approach's flexibility paves the way for research into hybrid federated learning models that adapt to changing client environments and data. Future studies will focus on integrating personalized learning paths to enhance model adaptability and efficiency across various datasets.

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

## Appendix

## A  Convergence Proof

### A.1  Convergence Analysis for Convex Setting

For the convex setting, we first start with the local training at client side using gradient descent. The update rule for gradient descent is presented in Eq. 17.

$$w_{t+1} = w_t - \eta_t \nabla F(w_t), \tag{17}$$

where $\eta_t$ is the learning rate. Using the smoothness property of $F(w)$, we expand $F(w_{t+1})$ as follows:

$$F(w_{t+1}) \leq F(w_t) + \langle \nabla F(w_t), w_{t+1} - w_t \rangle + \frac{L}{2} \|w_{t+1} - w_t\|^2. \tag{18}$$

Substituting $w_{t+1} - w_t = -\eta_t \nabla F(w_t)$ in inequality 18, we get inequality 19:

$$F(w_{t+1}) \leq F(w_t) - \eta_t \|\nabla F(w_t)\|^2 + \frac{L\eta_t^2}{2} \|\nabla F(w_t)\|^2. \tag{19}$$

Taking expectations of inequality 19 we get inequality 20:

$$\mathbb{E}[F(w_{t+1})] \leq \mathbb{E}[F(w_t)] - \eta_t \mathbb{E}[\|\nabla F(w_t)\|^2] + \frac{L\eta_t^2}{2} \mathbb{E}[\|\nabla F(w_t)\|^2]. \tag{20}$$

Rearranging inequality 20, we get inequality 21.

$$\eta_t \mathbb{E}[\|\nabla F(w_t)\|^2] \leq \mathbb{E}[F(w_t)] - \mathbb{E}[F(w_{t+1})] + \frac{L\eta_t^2}{2} \mathbb{E}[\|\nabla F(w_t)\|^2]. \tag{21}$$

Summing over $t = 0$ to $T - 1$ in the inequality 21, we get inequality 22:

$$\sum_{t=0}^{T-1} \eta_t \mathbb{E}[\|\nabla F(w_t)\|^2] \leq F(w_0) - F(w_T) + \sum_{t=0}^{T-1} \frac{L\eta_t^2}{2} \mathbb{E}[\|\nabla F(w_t)\|^2]. \tag{22}$$

Using the learning rate $\eta_t = \frac{\eta}{\sqrt{t+1}}$, the second term on the RHS of inequality 22 scales as $O(1/\sqrt{t+1})$. Dividing both sides by $\sum_{t=0}^{T-1} \eta_t$, we get inequality 23.

$$\frac{1}{\sum_{t=0}^{T-1} \eta_t} \sum_{t=0}^{T-1} \eta_t \mathbb{E}[\|\nabla F(w_t)\|^2] \leq \frac{F(w_0) - F(w_T)}{\sum_{t=0}^{T-1} \eta_t} + \frac{\sum_{t=0}^{T-1} \frac{L\eta_t^2}{2}}{\sum_{t=0}^{T-1} \eta_t}. \tag{23}$$

Let's analyze the left-hand side of the Eq. 23. By convexity of $F(w)$ and Jensen's inequality, we have:

$$\mathbb{E}[\|\nabla F(\bar{w}_T)\|^2] \leq \frac{1}{\sum_{t=0}^{T-1} \eta_t} \sum_{t=0}^{T-1} \eta_t \mathbb{E}[\|\nabla F(w_t)\|^2], \tag{24}$$

where $\bar{w}_T = \frac{1}{\sum_{t=0}^{T-1} \eta_t} \sum_{t=0}^{T-1} \eta_t w_t$ is a weighted average of iterates. For the learning rate schedule $\eta_t = \frac{\eta}{\sqrt{t+1}}$, we can establish that:

$$\sum_{t=0}^{T-1} \eta_t = \sum_{t=0}^{T-1} \frac{\eta}{\sqrt{t+1}} \approx 2\eta\sqrt{T}. \tag{25}$$

For the second term on the right-hand side:

$$\frac{\sum_{t=0}^{T-1} \frac{L\eta_t^2}{2}}{\sum_{t=0}^{T-1} \eta_t} = \frac{L}{2} \frac{\sum_{t=0}^{T-1} \frac{\eta^2}{t+1}}{\sum_{t=0}^{T-1} \frac{\eta}{\sqrt{t+1}}} \approx \frac{L\eta^2 \log(T)}{4\eta\sqrt{T}} = \frac{L\eta \log(T)}{4\sqrt{T}} \tag{26}$$

Using bound on $F(w_T) - F(w^*)$ where $w^*$ is the optimal solution, and substituting these results into Equation 23:

$$\mathbb{E}[\|\nabla F(\bar{w}_T)\|^2] \leq \frac{F(w_0) - F(w^*)}{2\eta\sqrt{T}} + \frac{L\eta \log(T)}{4\sqrt{T}} = \frac{1}{\sqrt{T}} \left( \frac{F(w_0) - F(w^*)}{2\eta} + \frac{L\eta \log(T)}{4} \right) \tag{27}$$

Setting $C = \frac{F(w_0) - F(w^*)}{2\eta} + \frac{L\eta \log(T)}{4}$, we obtain Eq. 28:

$$\mathbb{E}[F(w_T) - F(w^*)] \leq \frac{C}{\sqrt{T}}, \tag{28}$$

where $C$ is a constant that depends on $L$, $\sigma$, and the initial distance $\|w_0 - w^*\|^2$. The global objective function is defined as:

$$F(w) = \frac{1}{K} \sum_{i=1}^{K} f_i(w),$$

where $f_i(w)$ is the loss function for the client $i$. Using the smoothness property, we get inequality 29.

$$F(w_{t+1}) \leq F(w_t) + \langle \nabla F(w_t), w_{t+1} - w_t \rangle + \frac{L}{2} \|w_{t+1} - w_t\|^2. \tag{29}$$

Substitute $w_{t+1} - w_t = -\eta_t \nabla F(w_t)$ in the above inequality 29, we have inequality 30:

$$F(w_{t+1}) \leq F(w_t) - \eta_t \|\nabla F(w_t)\|^2 + \frac{L\eta_t^2}{2} \|\nabla F(w_t)\|^2. \tag{30}$$

Taking expectations of inequality 30 we get inequality 31:

$$\mathbb{E}[F(w_{t+1})] \leq \mathbb{E}[F(w_t)] - \eta_t \mathbb{E}[\|\nabla F(w_t)\|^2] + \frac{L\eta_t^2}{2} \mathbb{E}[\|\nabla F(w_t)\|^2]. \tag{31}$$

Rearranging the above inequality 31, we have inequality 32 as:

$$\eta_t \mathbb{E}[\|\nabla F(w_t)\|^2] \leq \mathbb{E}[F(w_t)] - \mathbb{E}[F(w_{t+1})] + \frac{L\eta_t^2}{2} \mathbb{E}[\|\nabla F(w_t)\|^2]. \tag{32}$$

Simplifying the above inequality, we get:

$$\mathbb{E}[F(w_t) - F(w_{t+1})] \geq \eta_t \mathbb{E}[\|\nabla F(w_t)\|^2] - \frac{L\eta_t^2}{2} \mathbb{E}[\|\nabla F(w_t)\|^2].$$

Let $\Delta_t$ denote the gradient noise. Using the bounded variance assumption:

$$\mathbb{E}[\|\Delta_t\|^2] \leq \sigma^2,$$

The overall gradient norm is bounded as:

$$\mathbb{E}[\|\nabla F(w_t)\|^2] \leq G^2 + \frac{\sigma^2}{K},$$

where $G^2$ is a bound on the true gradient norm. Incorporating the gradient variance bounds, we get the below inequality 33.

$$\mathbb{E}[F(w_t) - F(w_{t+1})] \geq \eta_t \mathbb{E}[\|\nabla F(w_t)\|^2] - \frac{L\eta_t^2}{2} \|\Delta_t\|^2. \tag{33}$$

Substitute $\|\Delta_t\|^2 \leq G^2$ in the above inequality, we have:

$$\mathbb{E}[F(w_t) - F(w_{t+1})] \geq \eta_t \mathbb{E}[\|\nabla F(w_t)\|^2] - \frac{L\eta_t^2 G^2}{2}.$$

Expressing this equivalently, we get inequality 34 and 35:

$$\mathbb{E}[F(w_{t+1}) - F(w_t)] \leq -\eta_t \mathbb{E}\|\nabla F(w_t)\|^2 + \frac{L\eta_t^2 G^2}{2} \tag{34}$$

$$\mathbb{E}[F(w_{t+1}) - F(w_t)] \leq -\eta_t \mathbb{E}\|\nabla F(w_t)\|^2 + \frac{L\eta_t^2}{2} \mathbb{E}[\|\Delta_t\|^2]. \tag{35}$$

So we have bounded $\mathbb{E}\|\Delta_t\|^2 \leq 2(\lambda_{\max}L_{KL})^2 + 2\sigma^2/K$. Let us denote $G^2 = 2(\lambda_{\max}L_{KL})^2 + 2\sigma^2/K$ for simplicity, where $\lambda_{\max}$ represents the largest eigenvalue of the Hessian matrix of the loss function and $L_{KL}$ represents the Lipschitz smoothness constant associated with the KL divergence term used in our adaptive loss function.

Now, by the convexity of $F$, we have the inequality 36.

$$F(w_t) - F(w^*) \leq \langle \nabla F(w_t), w_t - w^* \rangle \tag{36}$$

Rearranging inequality 34, 35 and using this convexity property defined in inequality 36, we achieve the following:

$$\eta_t \|\nabla F(w_t)\|^2 \leq F(w_t) - \mathbb{E}[F(w_{t+1})] + \frac{L\eta_t^2 G^2}{2}$$

$$\langle \nabla F(w_t), w_t - w^* \rangle \leq \frac{\|w_t - w^*\|^2 - \mathbb{E}[\|w_{t+1} - w^*\|^2]}{2\eta_t} + \frac{\eta_t}{2}\|\nabla F(w_t)\|^2$$

$$F(w_t) - F(w^*) \leq \frac{\|w_t - w^*\|^2 - \mathbb{E}[\|w_{t+1} - w^*\|^2]}{2\eta_t} + \frac{F(w_t) - \mathbb{E}[F(w_{t+1})]}{2} + \frac{L\eta_t G^2}{4}$$

Summing over $t = 0$ to $T - 1$ and using telescoping sums:

$$\sum_{t=0}^{T-1}(F(w_t) - F(w^*)) \leq \frac{\|w_0 - w^*\|^2}{2\eta_0} + \sum_{t=1}^{T-1}\|w_t - w^*\|^2\left(\frac{1}{2\eta_t} - \frac{1}{2\eta_{t-1}}\right)$$

$$+ \frac{F(w_0) - \mathbb{E}[F(w_T)]}{2} + \frac{LG^2}{4}\sum_{t=0}^{T-1}\eta_t$$

Choosing $\eta_t = \frac{\eta}{\sqrt{t+1}}$ where $\eta > 0$ is a constant, we have:

$$\sum_{t=0}^{T-1}\eta_t \leq 2\eta\sqrt{T}$$

Using Jensen's inequality and the convexity of $F$:

$$F(\bar{w}_T) - F(w^*) \leq \frac{1}{T}\sum_{t=0}^{T-1}(F(w_t) - F(w^*)).$$

Combining these results:

$$F(\bar{w}_T) - F(w^*) \leq \frac{\|w_0 - w^*\|^2}{2\eta T\sqrt{T}} + \frac{F(w_0) - F(w^*)}{2T} + \frac{L\eta G^2\sqrt{T}}{2T}.$$

This is the convergence rate for the convex setting of the proposed FedDUAL method.

### A.2 Convergence Analysis for Non-Convex Setting

From the smoothness assumption, we have below inequality 37:

$$F(w_{t+1}) \leq F(w_t) - \eta_t\|\nabla F(w_t)\|^2 + \frac{L\eta_t^2}{2}\|\nabla F(w_t)\|^2. \tag{37}$$

Taking expectations of inequality 37, we get 38:

$$\mathbb{E}[F(w_{t+1})] \leq \mathbb{E}[F(w_t)] - \eta_t \mathbb{E}[\|\nabla F(w_t)\|^2] + \frac{L\eta_t^2}{2}\mathbb{E}[\|\nabla F(w_t)\|^2]. \tag{38}$$

Rearranging the above inequality 38, we get inequality 39:

$$\eta_t \mathbb{E}[\|\nabla F(w_t)\|^2] \leq \mathbb{E}[F(w_t)] - \mathbb{E}[F(w_{t+1})] + \frac{L\eta_t^2}{2}\mathbb{E}[\|\nabla F(w_t)\|^2]. \tag{39}$$

Summing over $t = 0$ to $T - 1$ in the inequality 39, we get 40:

$$\sum_{t=0}^{T-1} \eta_t \mathbb{E}[\|\nabla F(w_t)\|^2] \leq F(w_0) - F(w_T) + \sum_{t=0}^{T-1} \frac{L\eta_t^2}{2}\mathbb{E}[\|\nabla F(w_t)\|^2]. \tag{40}$$

Assuming a constant learning rate $\eta_t = \eta$ and substituting it into inequality 40, we obtain 40:

$$\sum_{t=0}^{T-1} \eta \mathbb{E}[\|\nabla F(w_t)\|^2] \leq F(w_0) - F(w_T) + \frac{L\eta^2 T}{2}. \tag{41}$$

Rearranging inequality 41 yields 42.

$$\frac{1}{T}\sum_{t=0}^{T-1} \mathbb{E}[\|\nabla F(w_t)\|^2] \leq \frac{F(w_0) - F(w_T)}{\eta T} + \frac{L\eta}{2}. \tag{42}$$

With $\eta$ chosen as $\sqrt{\frac{2(F(w_0)-F(w^*))}{LG^2 T}}$ to balance the terms, substituting it into inequality 42 results the following result:

$$\frac{1}{T}\sum_{t=0}^{T-1} \mathbb{E}[\|\nabla F(w_t)\|^2] \leq C\sqrt{\frac{1}{T}}, \tag{43}$$

where $C$ is a constant depending on $F(w_0)$, $F(w^*)$, $L$, and $G^2$. We proceed with the same inequality as established in the convex setting:

$$\mathbb{E}[F(w_{t+1}) - F(w_t)] \leq -\eta_t \|\nabla F(w_t)\|^2 + \frac{L\eta_t^2 G^2}{2} \tag{44}$$

Summing over $t = 0$ to $T - 1$ in inequality 44, we obtain the following inequality:

$$\sum_{t=0}^{T-1} \mathbb{E}[F(w_{t+1}) - F(w_t)] \leq -\sum_{t=0}^{T-1} \eta_t \|\nabla F(w_t)\|^2 + \frac{LG^2}{2}\sum_{t=0}^{T-1} \eta_t^2$$

$$F(w^*) - F(w_0) \leq -\sum_{t=0}^{T-1} \eta_t \|\nabla F(w_t)\|^2 + \frac{LG^2}{2}\sum_{t=0}^{T-1} \eta_t^2.$$

Using the learning rate $\eta_t = \frac{\eta}{\sqrt{T}}$ (constant for all $t$), we have:

$$F(w^*) - F(w_0) \leq -\frac{\eta}{\sqrt{T}}\sum_{t=0}^{T-1} \|\nabla F(w_t)\|^2 + \frac{LG^2 \eta^2}{2}$$

$$\frac{1}{T}\sum_{t=0}^{T-1} \|\nabla F(w_t)\|^2 \leq \frac{\sqrt{T}}{\eta T}(F(w_0) - F(w^*)) + \frac{LG^2 \eta}{2\sqrt{T}}.$$

By choosing $\eta = \sqrt{\frac{2(F(w_0) - F(w^*))}{LG^2}}$ to balance the terms, we obtain:

$$\frac{1}{T} \sum_{t=0}^{T-1} \mathbb{E}\|\nabla F(w_t)\|^2 \leq \sqrt{\frac{2LG^2(F(w_0) - F(w^*))}{T}}$$

This yields the required convergence rate for the non-convex setting of the proposed FedDUAL method.

## B    Discussion on convergence rates

### B.1    Convex Setting

In the convex setting, The rate is optimal for first-order stochastic methods in convex optimization. The key points are:

1. The result is for the average iterate $\bar{w}_T$, which is common in convex stochastic optimization.
2. The rate depends on the initial distance to the optimum $\|w_0 - w^*\|$, the initial suboptimality $F(w_0) - F(w^*)$, and the bound on the stochastic gradients $G^2$.

### B.2    Non-Convex Setting

For the non-convex setting, The result is significantly different from the convex case:

1. We measure convergence in terms of the gradient norm, not the function value, as finding global minima is generally intractable in non-convex settings.
2. The result is for the average of the squared gradient norms, not the minimum.
3. This rate is also optimal for first-order stochastic methods in non-convex smooth optimization.

### B.3    Comparison and Implications

1. Different Metrics: The convex and non-convex results use different metrics for convergence, which is standard in optimization theory. In the convex case, we bound the suboptimality of the function value, while in the non-convex case, we bound the average squared gradient norm.

2. Similarity in Rates: Both settings achieve an $O(1/\sqrt{T})$ rate, but the meaning is different. In the convex case, this rate applies to the optimality gap, while in the non-convex case, it applies to the average squared gradient norm.

3. Adaptive Loss and Wasserstein Barycenter: These novel components of our algorithm are captured in the constant factors and the bound $G^2$ on the stochastic gradients. They do not affect the asymptotic rate but improves the practical performance by reducing client heterogeneity.

4. Communication Efficiency: The $O(1/\sqrt{T})$ rate in both cases is in terms of the number of communication rounds, suggesting that our method is communication-efficient in both convex and non-convex settings.

5. Practical Implications: While the asymptotic rates are the same, the non-convex case may require more iterations to achieve a desired level of accuracy, as we're working with squared gradient norms rather than function values.

In conclusion, our analysis differentiates between the convex and non-convex settings, providing appropriate convergence guarantees for each case. The proposed federated learning algorithm achieves optimal rates in both settings, demonstrating its theoretical efficiency and robustness to the convexity of the objective function. We achieve a convergence rate of $O\left(\frac{1}{\sqrt{T}}\right)$ towards a stationary point under smooth, non-convex conditions. This rate is on par with the traditional FedAvg method in non-convex settings (Yu et al., 2019).

Table 3: Ablation study results for the proposed FedDUAL method on CIFAR10 dataset.

| Adaptive Loss | Dynamic Aggregation | Accuracy (%) |
|:---:|:---:|:---:|
| ✓ | ✗ | $41.05 \pm 0.12$ |
| ✗ | ✓ | $46.50 \pm 0.15$ |
| ✓ | ✓ | $\mathbf{48.70 \pm 0.20}$ |

Table 4: Ablation study results for the proposed FedDUAL method on CIFAR100 dataset.

| Adaptive Loss | Dynamic Aggregation | Accuracy (%) |
|:---:|:---:|:---:|
| ✓ | ✗ | $25.05 \pm 0.11$ |
| ✗ | ✓ | $27.01 \pm 0.17$ |
| ✓ | ✓ | $\mathbf{29.15 \pm 0.24}$ |

### B.4  Lower Bounds and Optimality Discussion

In stochastic convex optimization with smooth functions, the minimax lower bound on the expected optimality gap is well-established as: $\Omega(1/\sqrt{T})$ (Nemirovskij & Yudin, 1983), (Agarwal et al., 2012). Specifically, for any first-order stochastic algorithm $\mathcal{A}$ on the class of $L$-smooth convex functions $\mathcal{F}$, there exists a function $f \in \mathcal{F}$ such that:

$$\mathbb{E}[f(\bar{w}_T) - f(w^*)] \geq \frac{c\sigma D}{\sqrt{T}} \tag{45}$$

where $c > 0$ is a universal constant, $\sigma$ is the bound on the stochastic gradient variance, and $D = \|w_0 - w^*\|$ is the initial distance to the optimum.

For non-convex smooth optimization, Arjevani et al. (2019) established that the lower bound for the minimum expected squared gradient norm is $\Omega(1/\sqrt{T})$. Formally, for any first-order stochastic algorithm $\mathcal{A}$ on the class of $L$-smooth non-convex functions $\mathcal{F}$, there exists a function $f \in \mathcal{F}$ such that:

$$\min_{t \in \{0,1,\ldots,T-1\}} \mathbb{E}[\|\nabla f(w_t)\|^2] \geq \frac{c'L\Delta_f\sigma^2}{T} \tag{46}$$

where $c' > 0$ is a universal constant, $\Delta_f = f(w_0) - \inf_w f(w)$ is the initial function value gap, and $\sigma$ is the bound on the stochastic gradient variance.

The convergence rates for FedDUAL match these lower bounds up to constant factors, achieving $O(1/\sqrt{T})$ for the convex case and $O(1/\sqrt{T})$ for the non-convex case. This indicates that the proposed method attains the optimal dependency on $T$, which is the number of communication rounds. While FedDUAL shares the asymptotic convergence rates of standard federated methods like FedAvg (McMahan et al., 2017) and FedProx (Li et al., 2020), its key advantage lies in effectively mitigating heterogeneity. This is evident in the constant factors of the convergence bounds, which are influenced by client drift—an issue FedDUAL addresses through dynamic loss adaptation and Wasserstein barycenter aggregation.

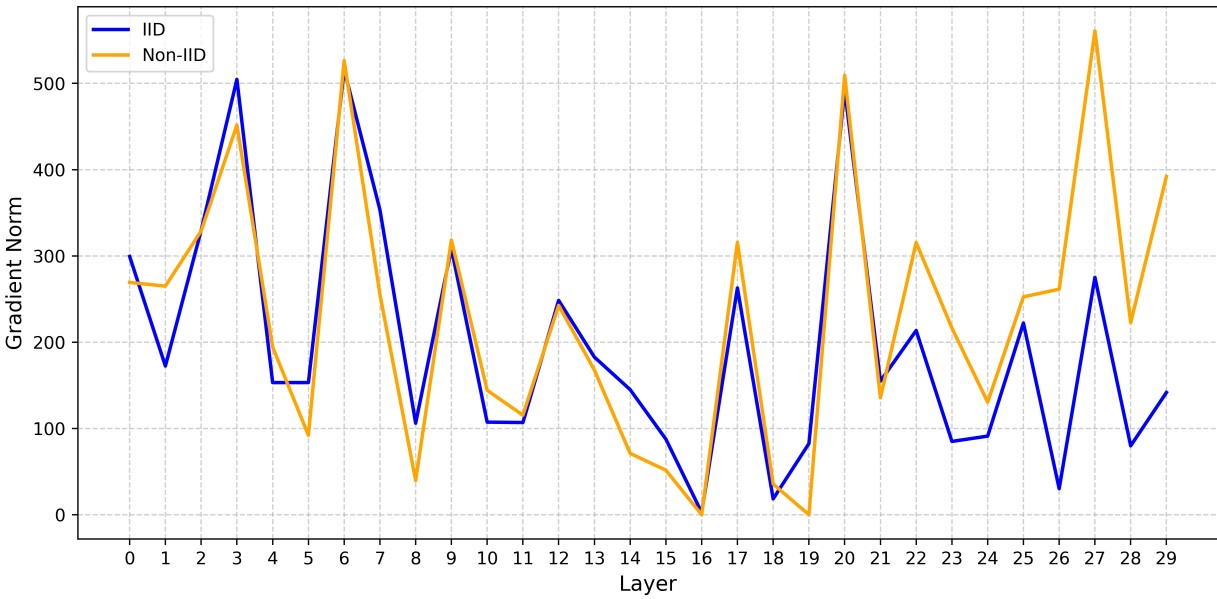

Figure 6: Comparison of gradient norms between models trained on IID and non-IID datasets using the FedAvg algorithm on the CIFAR10 dataset using VGG16 model.

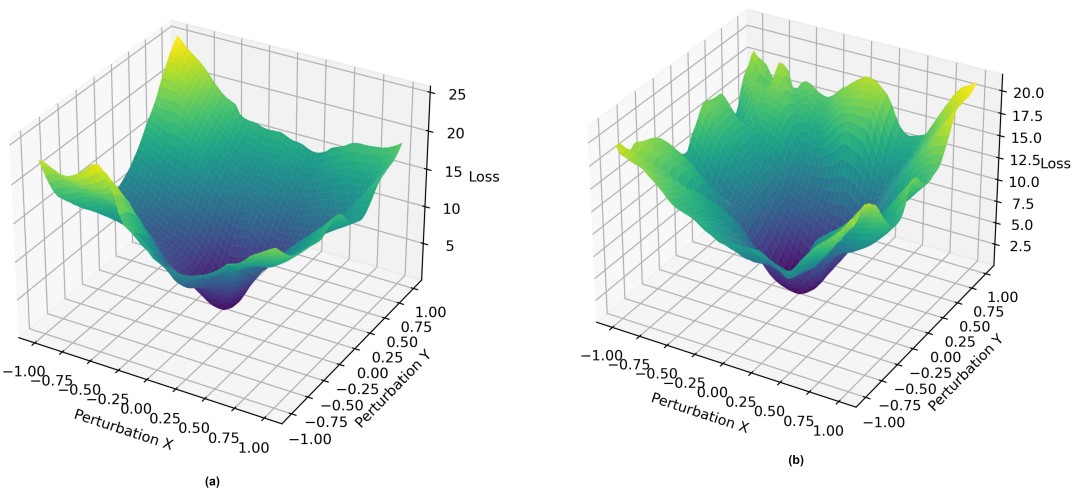

Figure 7: Visualization of the loss surface for the global model trained on the FMNIST dataset with non-IID data ($\alpha = 0.01$): (a) shows the loss surface for the global model trained using FedAvg, while (b) depicts the loss surface for the global model trained with the proposed method FedDUAL.

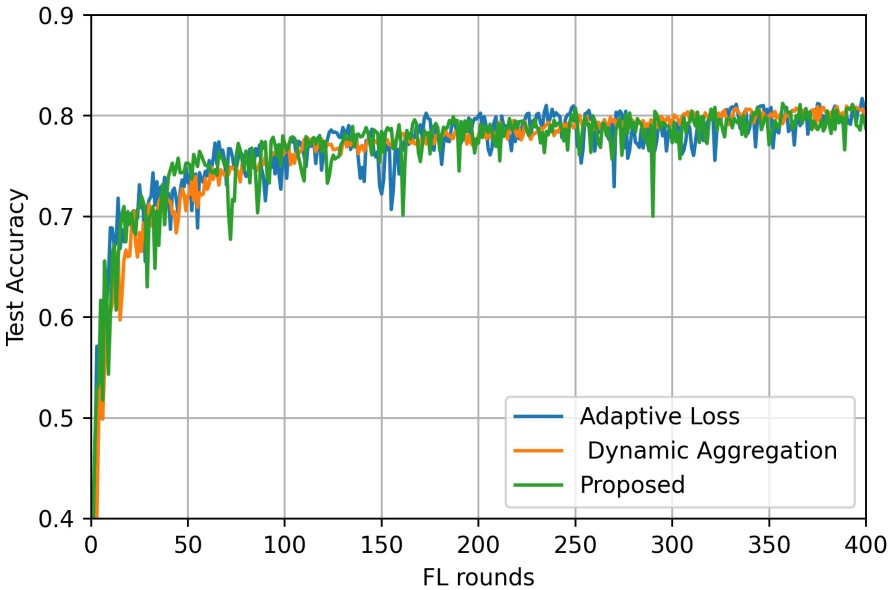

Figure 8: Learning curves of the individual modules and the proposed method.

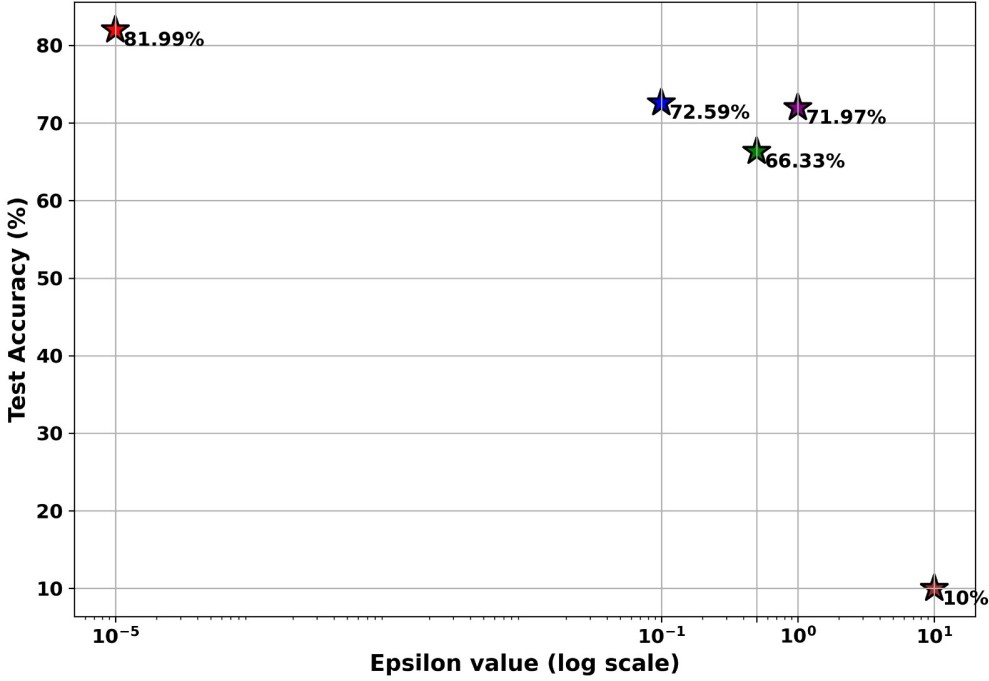

Figure 9: Performance of the proposed method with different epsilon values.

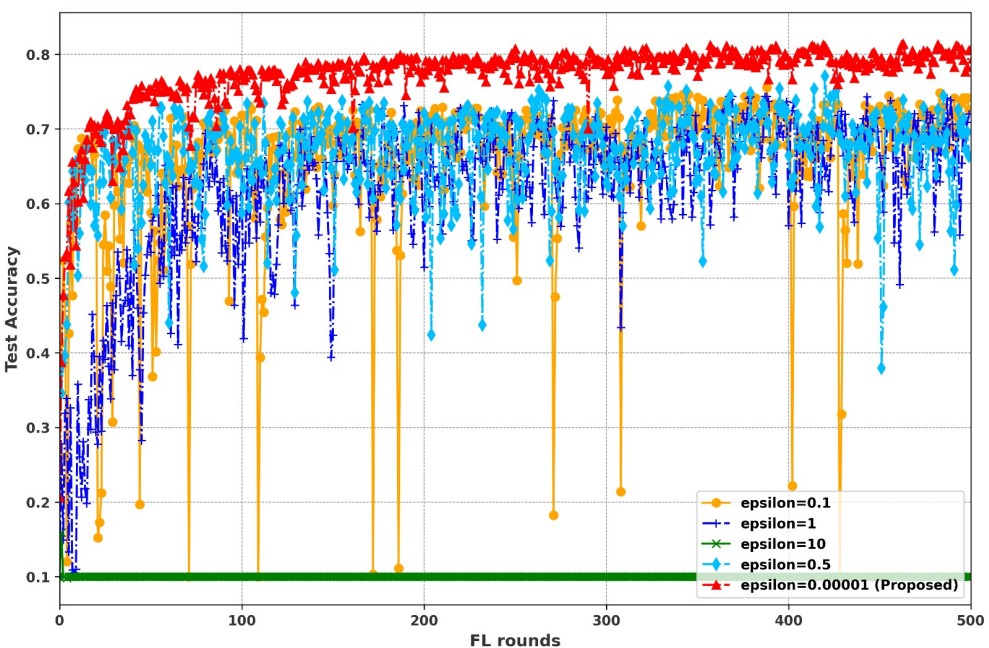

Figure 10: Learning curve of the proposed method with different epsilon values.

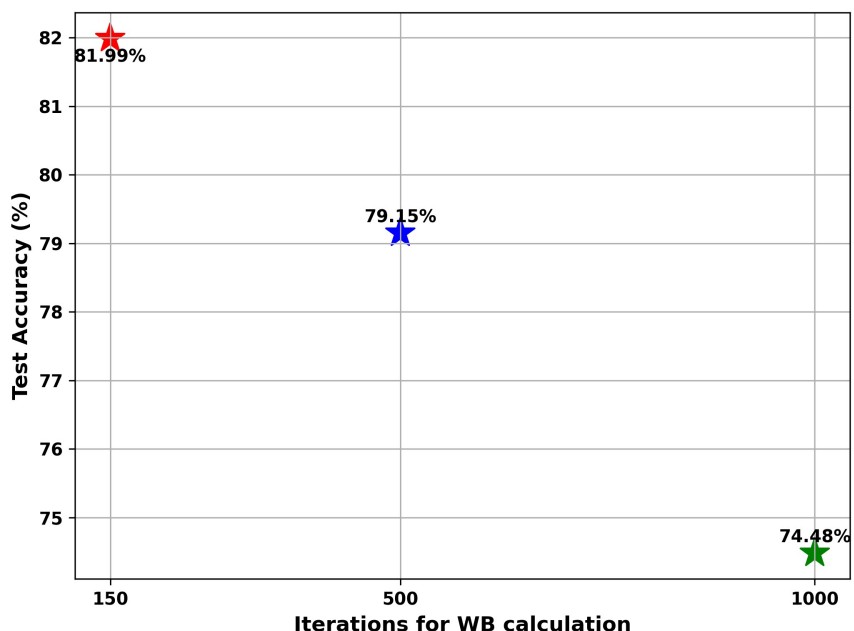

Figure 11: Performance across different number of Iterations for WB calculation.

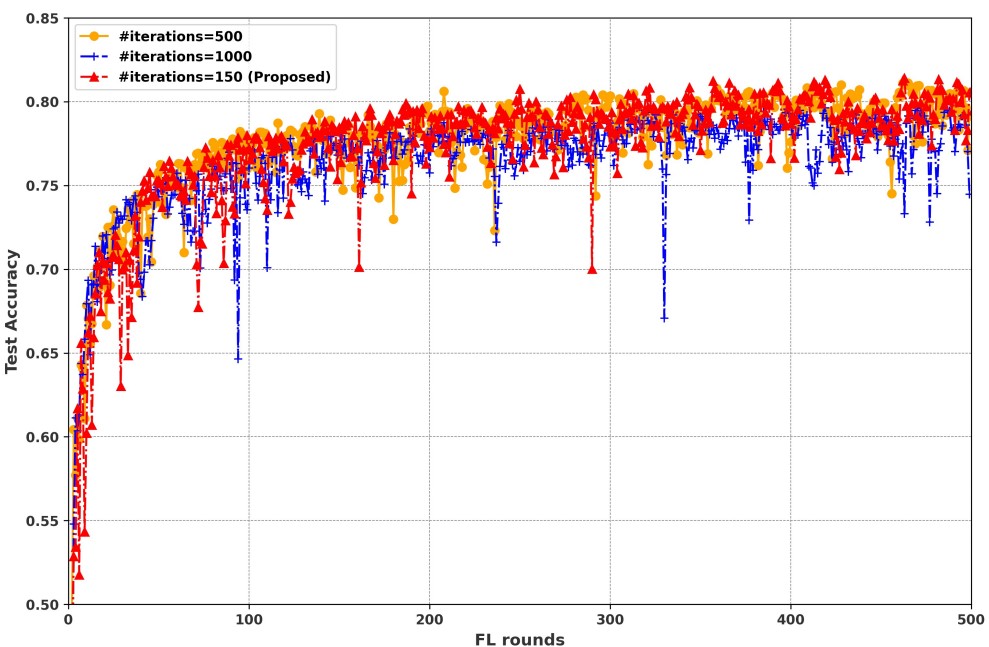

Figure 12: Learning curves for different number of Iterations for WB calculation.

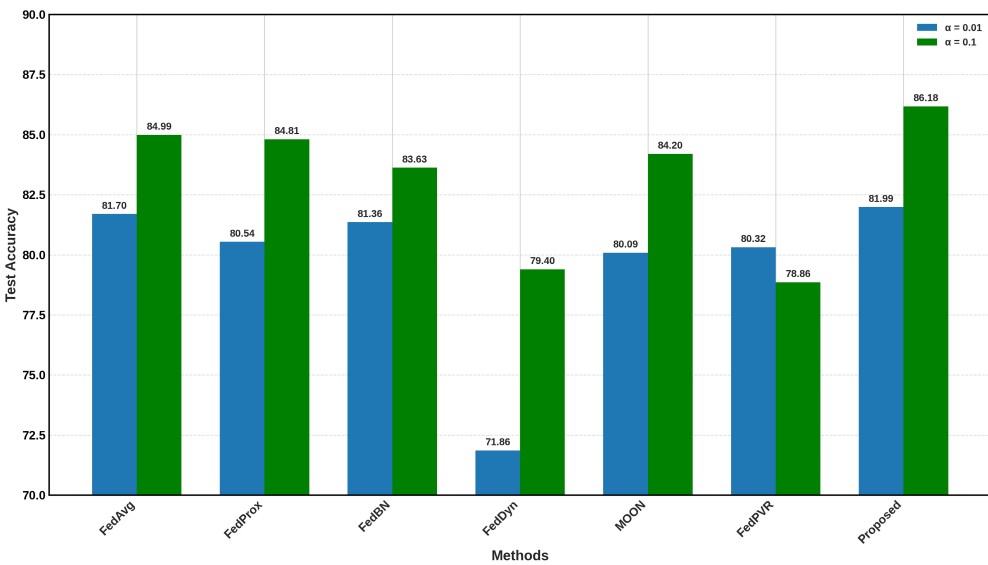

Figure 13: Illustrates the accuracy of the proposed method and baselines across different levels of data heterogeneity on the FMNIST dataset.

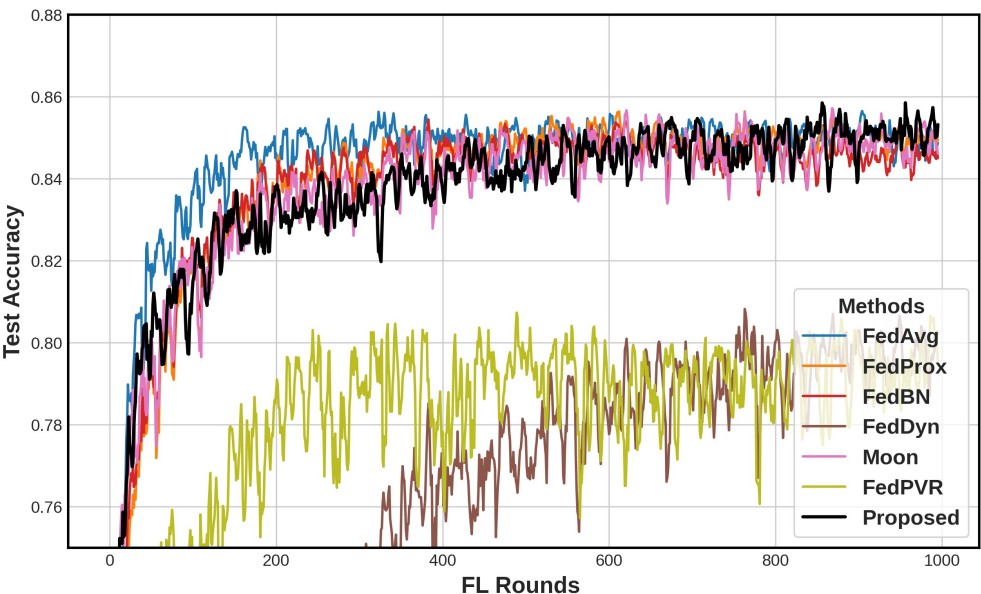

Figure 14: Learning curve of the proposed method and other baselines on FMNIST dataset with data heterogeneity level $\alpha$=0.1.

