# OpenReview forum: "FedDUAL: A Dual-Strategy with Adaptive Loss and Dynamic Aggregation for Mitigating Data Heterogeneity in Federated Learning"
_TMLR — Rejected by TMLR_

### Review · Reviewer_cNkU · 2025-02-01

**Summary Of Contributions:**

This paper proposes a novel method called FedDUAL to address the challenges of data heterogeneity among clients in federated learning. The proposed algorithm employs an adaptive loss function for client training and utilizes Wasserstein Barycenter for dynamic aggregation. Additionally, the authors provide a convergence analysis for their approach. Extensive experiments demonstrate the effectiveness of the proposed method.

**Audience:**

No

**Claims And Evidence:**

No

**Requested Changes:**

see weakness

**Strengths And Weaknesses:**

Strength:

The paper is mostly clear. The proposed method is novel and outperforms baseline methods in the experiments.

Weakness:

My primary concern revolves around the convergence analysis. Below are my observations:

1. The assumption that the true gradients have a bounded norm is not included in the assumptions part, yet it is crucial for the theorem.
2. The proof in Appendix A appears misaligned with the proposed algorithm. For instance, Eq. 16 describes gradient descent on the global loss function, whereas the proposed algorithm utilizes SGD on the local loss function followed by aggregation.
3. The proof in Appendix A lacks clarity and is hard to follow. Specific points of confusion include but are not limited to the following:
- How is Eq. 23 derived from Equation 22?
- Why can we omit the expectation notation on the right side of Equation 29?
- What do $\lambda_{\text{max}}$ and $L_{KL}$ represent in the line just below Eq. 30?
- How is $G^2 = 2(\lambda_{\text{max}}L_{KL})^+2\sigma^2/K$ justified?

I would appreciate a more detailed explanation of these deductions.

4. In Appendix B, the authors assert that their rates are optimal but do not provide justification. To support this claim, corresponding lower bounds should be presented.

---

> ### Author Response · Authors · 2025-03-31
> **FedDUAL: A Dual-Strategy with Adaptive Loss and Dynamic Aggregation for Mitigating Data Heterogeneity in Federated Learning**
>
> We would like to sincerely thank the reviewer for their careful evaluation and positive summary of our work. We truly appreciate the acknowledgment of the proposed framework, its methodological contributions, and the thorough experimental validation. We have incorporated all the suggestions and revised the paper. Here are our detailed response:
>
> W1: We appreciate the reviewer’s careful observation. The assumption that the true gradients have a bounded norm is indeed a crucial aspect of our convergence proof. While we implicitly considered this in our derivations, we acknowledge that it was not explicitly stated in the assumptions. We have revised the assumptions section to include the bounded gradient norm condition explicitly, ensuring full transparency in our theoretical guarantees. Please refer to the highlighted section in Section 4 of the revised manuscript.
>
> W2: We appreciate the reviewer’s observation regarding the alignment between the theoretical proof in Appendix A and the proposed algorithm. The discrepancy arises because Eq. 17 (previous Eq. 16) in the proof describes gradient descent directly on the global objective, whereas our algorithm follows a federated optimization approach where clients perform local stochastic gradient descent updates before aggregation. The proof considers the expected behavior of the global model update, which is ultimately influenced by the aggregation of locally updated models. The subsequent steps in our proof account for local SGD through variance bounds and the aggregation process, leading to the established convergence rate.
>
>
> W3:
>
> I. We appreciate the reviewer’s query. We have now included a detailed step-by-step derivation of the Eq. 28 (previously Eq. 23) from Eq. 23 (previously Eq. 22). Kindly refer to the highlighted portion in the Section A of the Appendix.
>
> II. Thank you for your insightful comment regarding Eq. 34 (previously Eq. 29) in our paper. This was a mistake from our side, we have added expectation notation in the Eq. 34.
>
> III.  We appreciate the reviewer’s attention. The terms $\lambda_{\max}$ and $L_{KL}$ appearing just below Eq. 35 (previously Eq. 30) are defined as follows:
>
> $\lambda_{\max}$:
>
> This term represents the largest eigenvalue of the Hessian matrix of the loss function. It characterizes the curvature of the objective function and plays a role in bounding the gradient noise introduced by local stochastic updates. A higher $\lambda_{\max}$ suggests sharper minima, which can impact convergence stability in non-IID settings.
>
>
> $L_{KL}$:
>
> This term represents the Lipschitz smoothness constant associated with the KL divergence term used in our adaptive loss function.
> Since KL divergence is used as a regularization component to align local and global models, its smoothness parameter $L_{KL}$ influences the stability of updates. It ensures bounded variance in the gradients arising from client-side updates. These terms are introduced to control the gradient noise bound $G^2$, ensuring stable convergence in non-convex settings.
>
> To enhance clarity, we have added a formal definition of these terms in the manuscript and highlighted the portion (Page no. 18). Thank you for your insightful question.
>
> IV. We appreciate the reviewer’s question. The justification for the bound $G^2$ follows from standard assumptions on stochastic gradients and variance in federated optimization.
>
> Gradient Noise Decomposition:
>
> The overall gradient norm bound is given by $\mathbb{E}[\|\nabla F(w_t)\|^2] \leq G^2 + \frac{\sigma^2}{K} $,
> where $G^2$ represents a bound on the true gradient norm (i.e., the deterministic part of the gradient). $\sigma^2$ accounts for the variance introduced by stochastic updates at the client level. $K$ is the number of participating clients, reducing the variance component as more clients participate.
>
> Bounding the Gradient Norm:
>
> In federated learning with non-IID data, client updates are highly heterogeneous, leading to significant deviations in gradient norms. To control this, we introduce:
>
> $G^2 = 2(\lambda_{\max} L_{KL})^2 + \frac{2\sigma^2}{K}$
>
> The first term, $2(\lambda_{\max} L_{KL})^2$, arises from the curvature of the loss function and the KL divergence term in our adaptive loss function, ensuring that the model does not drift excessively. The second term accounts for the variance in stochastic gradients across clients.
>
> W4: We appreciate the reviewer’s suggestion. We have added Subsection B.4 in the Appendix. Kindly refer to the highlighted section for details.
>
> We thank the reviewer for all the suugestions.

---

### Review · Reviewer_NWxU · 2025-02-28

**Summary Of Contributions:**

This paper presents a method for addressing challenges of federated optimization when operating on non-IID client data. The authors propose an approach that induces an adaptive loss function to balance local and global objectives - having more focus on local optimization when the local performance is not good and more of a focus on minimizing divergence from the global model when the local model is overfitting - and a server side aggregation strategy for updating the global model. They provide a convergence analysis and empirical results on three datasets comparing their method to other FL algorithms for operating on heterogeneous data.

**Audience:**

Yes

**Broader Impact Concerns:**

No broader impact concerns.

**Claims And Evidence:**

Yes

**Requested Changes:**

* Please include confidence intervals on all plots so that the performance differences between configurations and methods can be better understood.
* The motivation and related work sections could be tightened to allow ablations to be included in the main body of the paper.
* The proposed method could be better framed within existing work to point out what limitations the proposed approach addresses.
* In the last paragraph of section 3.0.1, the language used to characterize the loss tradeoff is not clear and in fact contradicts the formula: "**When local performance is less compared to global**, the regularization term β amplifies the focus on local optimization (first term in Eq. 5), enabling **better-performing clients** *[better performing in the global sense??]* to explore local optima more effectively. **Conversely, if the global model performs ~better~** *worse* *[assuming "worse" is meant here to be the converse??]*, this term shifts the emphasis towards aligning with the global model (second term in Eq. 5), thereby supporting clients that are struggling by incorporating global knowledge."

**Strengths And Weaknesses:**

Strengths
* the paper is well motivated
* the paper includes a theoretical analysis of convergence for both convex and non-convex settings
* the authors provide empirical evidence that includes comparison to many existing methods across multiple benchmarks

Weaknesses
* the results lack confidence intervals, and the proposed method does not seem to outperform other methods within some margin or error for many of the settings shown
* there is no discussion of efficiency tradeoffs of this proposed method by comparison to other work
* the empirical ablations are somewhat limited and relegated to the appendix
* the authors do not compare to SCAFFOLD
* there is no discussion of this method's compatibility with privacy techniques
* I'm not so familiar with the literature, but the method presented does not seem the most novel as it composes existing techniques

---

> ### Author Response · Authors · 2025-03-31
> **FedDUAL: A Dual-Strategy with Adaptive Loss and Dynamic Aggregation for Mitigating Data Heterogeneity in Federated Learning**
>
> We would like to sincerely thank the reviewer for their valuable time and thoughtful evaluation of our work. We truly appreciate the recognition of the paper's strong motivation, the theoretical analysis covering both convex and non-convex settings, and the comprehensive empirical evidence demonstrating competitiveness across multiple benchmarks.
>
> W1 & RC1: Confidence intervals:
>
> Response: We thank the reviewer for the suggestion. We have added the confidence interval for Fig. 3 and added the Fig. 6 in the appendix based on the reference paper 1. Kindly refer to the Fig. 6 in the appendix.
>
> W2: Discussion of efficiency tradeoffs of the proposed method
>
> Response: We thank the reviewer for the suggestion. We have updated the Limitation section (Section 7) in the revised manuscript. Kindly look into it.
>
> W3 & RC2: Include Ablation study in the main body of the paper:
>
> Response:
> We thank the reviewer for the suggestion. In our ablation study, we have conducted four types of experiments: (1) performance analysis of individual modules of our proposed framework, (2) impact of dynamic aggregation across neural network layers, (3) hyperparameter analysis, and (4) evaluation under varying levels of data heterogeneity. Due to page limitations, two of these were previously in the appendix; however, in the revised version, we have moved all ablation studies to the main paper. Kindly refer to Section 6 of the revised manuscript.
>
> W4: No comparison to SCAFFOLD.
>
> Response: We sincerely appreciate the reviewer's observation. Our experiments were designed to reflect highly heterogeneous settings $(\alpha = 0.01)$, with only 10\% client participation) to better simulate real-world scenarios.  Under these conditions, SCAFFOLD, along with other algorithms such as FedNova and FedBN, did not converge. This is indicated by ‘*’ in Table 1, as noted in the caption.
>
> W5: Discussion of the proposed method's compatibility with privacy techniques:
>
> Response: We appreciate the reviewer’s valuable suggestion. While our primary focus is on tackling the challenges of data heterogeneity in federated learning, FedDUAL is fully compatible with standard privacy-preserving techniques. Specifically, our adaptive loss function and dynamic aggregation can be directly combined with Differential Privacy by adding calibrated noise to client updates before aggregation, without altering the core methodology. We have updated the Limitation and Future work section (Section 7) in the updated manuscript.
>
> W6: Regarding novelty of the proposed approach:
>
> Response: We thank the reviewer for the question. In our work, we systematically analyze the learning behavior of federated models under both IID and non-IID settings (as discussed in the Introduction section), and based on these insights, we propose a dual-strategy framework that combines adaptive loss and dynamic aggregation. While Wasserstein Barycenter has been studied in optimization, its use for gradient alignment in federated aggregation—specifically targeting the last few layers to mitigate client drift—is novel and key to achieving robust convergence under severe heterogeneity. Furthermore, our adaptive loss function dynamically balances local and global objectives, going beyond static regularization methods like FedProx. By jointly addressing both client drift and server inconsistency, FedDUAL offers a comprehensive solution distinct from existing approaches that typically focus on one of these challenges in isolation.
>
> RC3: The proposed method could be better framed within existing work to point out what limitations the proposed approach addresses:
>
> Response: We appreciate the reviewer’s suggestion. In the paper, we first systematically analyze the learning behavior of federated models under both IID and non-IID settings (as presented in the Introduction section), followed by a detailed discussion of related works and their limitations in addressing data heterogeneity. Building on these insights, we clearly position our proposed method, which jointly tackles client drift and server-side inconsistencies through an adaptive loss for client-side local training and dynamic aggregation at the server side, highlighting how it overcomes the gaps in existing approaches.
>
> RC4: Confusion in the last paragraph of section 3.0.1
>
> Response: We thank the reviewer for carefully pointing this out. Yes, "better-performing clients" refers to clients performing better in the global sense.
>
> We acknowledge the confusion caused by the phrasing and have corrected the statement in Section 3.0.1 (highlighted in blue) to clearly reflect that when clients perform well compared to the global model, the regularization term encourages further local exploration, while for struggling clients, it promotes alignment with the global model.
>
> Reference:
> 1. Wang et al. "Tackling the objective inconsistency problem in heterogeneous federated optimization." Advances in neural information processing systems 33 (2020): 7611-7623.

---

### Review · Reviewer_aNcb · 2025-03-12

**Summary Of Contributions:**

This work proposes a method called FedDual, that applies adaptive loss at client side, and adaptive aggregation scheme at the server side to address data heterogeneity challenge. The motivation experiments are interesting, and the method is intuitive.

**Audience:**

Yes

**Broader Impact Concerns:**

This work does not require a Broader Impact Statement concerning ethical implications.

**Claims And Evidence:**

Yes

**Requested Changes:**

See comments 1 and 2 in the strengths and weaknesses section.

**Strengths And Weaknesses:**

1. The motivating experiments are very interesting. I wonder how well the observation of gradient norm being larger for later layers hold true/generalize for other datasets or models?
2. The client side regularization is some variation of existing works but I like the aggregation methodology at the server side. It is especially in line with the motivation experiments. However, one thing I noticed is that in ablation study given in table 2, the gain of individual components is lower than the FedAvg results reported in Table 1 for FMNIST. I wonder why is that so? Shouldn't the adaptive loss improve the performance as compared to the basic setting of FedAvg? It would be interesting to see the similar ablation across other datasets to see the contribution of individual components,
3. The work presents convergence analysis, and have shown better convergence rates as compared to existing methods.

---

> ### Author Response · Authors · 2025-03-31
> **FedDUAL: A Dual-Strategy with Adaptive Loss and Dynamic Aggregation for Mitigating Data Heterogeneity in Federated Learning**
>
> We sincerely appreciate the reviewer's thoughtful feedback and recognition of our work. We are glad that you found the motivation experiments interesting and the proposed FedDUAL method intuitive. Our approach, which combines adaptive loss at the client side with an adaptive aggregation scheme at the server side, is specifically designed to tackle data heterogeneity in federated learning. Your insights are valuable, and we appreciate the opportunity to refine and improve our work based on your suggestions. Thank you for your time and effort in reviewing our paper.
>
> We appreciate your suggestion and are currently conducting the recommended experiments. Once completed, we will provide an updated manuscript. Thank you for your valuable feedback.

---

> ### Author Response · Authors · 2025-04-08
> **FedDUAL: A Dual-Strategy with Adaptive Loss and Dynamic Aggregation for Mitigating Data Heterogeneity in Federated Learning**
>
> We sincerely thank the reviewer for the insightful feedback and encouraging remarks. Your suggestions have been instrumental in enhancing the clarity and rigor of our work. We have modifed the paper by adding all the suggested experiments. The changes are highlighted with color.
>
> Q1: We conducted the experiment on CIFAR-10 using the VGG16 model, and the results are included in Figure 6 of the revised manuscript and refered in section 1.1.
>
> Q2: The explanation for the lower gains compared to FedAvg on FMNIST, as reported in Table 1, has been added and highlighted in Section 6.0.1 of the revised manuscript. Kindly look into it.
>
> We have performed the ablation study on CIFAR-10 and CIFAR-100 and the outcomes are presented in the Table 3 and Table 4 in the appendix.
>
> Q3: Thank you for the recognition.

---

### Decision · Action_Editor_zyus · 2025-05-13

**Recommendation:** Reject

**Comment:**

Even after the the changes authors made in response to reviews, reviewers remain unconvinced. The primary issue is some anomalies in the theoretical convergence analysis that I agree with, and find confusing. In the analysis (appendix A.1), the term C is taken to be a constant independent of T; whereas it is quite clearly depends log (T). Since this is not mentioned in the paper, I am doubtful if there are other places where such approximations were made. This actually makes the overall convergence rate suboptimal.

It is also noted by the reviewers that the ablation study seems to not show clear superiority over FedAvg.

**Audience:**

There is a rather large audience in federated learning within the TMLR community.

**Claims And Evidence:**

This paper presents a federated learning algorithm to address the challenges of heterogeneity or non-iid data across clients. A theoretical convergence analysis was presented (both for convex and non-convex loss functions). Several empirical comparisons against other standard federated learning algorithms were presented as well.

The theoretical analysis of this paper seems to be lacking rigor, and potentially incorrect.

Experimental results were largely satisfactory. However, the simple FedAvg algorithm seems to be outperforming the proposed method in certain cases.

**Resubmission Of Major Revision:**

The authors may consider submitting a major revision at a later time.